# Simple Sequence Repeat Markers Reveal Genetic Diversity and Population Structure of Bolivian Wild and Cultivated Tomatoes (*Solanum lycopersicum* L.)

**DOI:** 10.3390/genes13091505

**Published:** 2022-08-23

**Authors:** Evelyn E. Villanueva-Gutierrez, Eva Johansson, Maria Luisa Prieto-Linde, Alberto Centellas Quezada, Marie E. Olsson, Mulatu Geleta

**Affiliations:** 1Department of Plant Breeding, Swedish University of Agricultural Sciences, P.O. Box 190, SE-234 22 Lomma, Sweden; 2Departmento de Fitotecnia, Facultad de Ciencias Agricolas, Pecuarias y Forestales “Dr. Martín Cárdenas”, Universidad Mayor de San Simón, Cochabamba P.O. Box 4894, Bolivia

**Keywords:** core germplasm, AMOVA, UPGMA, landraces, advanced lines, modern cultivars

## Abstract

The western part of South America is a centre of diversity for tomatoes, but genetic diversity studies are lacking for parts of that region, including Bolivia. We used 11 simple sequence repeat (SSR) markers (including seven novel markers) to evaluate genetic diversity and population structure of 28 accessions (four modern cultivars, four advanced lines, nine landraces, 11 wild populations), and to compare their genetic variation against phenotypic traits, geographical origin and altitude. In total, 33 alleles were detected across all loci, with 2–5 alleles per locus. The top three informative SSRs were SLM6-11, LE20592 and TomSatX11-1, with polymorphism information content (PIC) of 0.65, 0.55 and 0.49, respectively. The genetic diversity of Bolivian tomatoes was low, as shown by mean expected heterozygosity (He) of 0.07. Analysis of molecular variance (AMOVA) revealed that 77.3% of the total variation was due to variation between accessions. Significant genetic differentiation was found for geographical origin, cultivation status, fruit shape, fruit size and growth type, each explaining 16–23% of the total variation. Unweighted Pair Group Method with Arithmetic Mean (UPGMA) tree and principal coordinate analysis (PCoA) scatter plot both revealed differentiation between accessions with determinate flowers and accessions with indeterminate flowers, regardless of cultivation status. The genetic profiles of the accessions suggest that the Bolivian tomato gene pool comprises both strictly self-pollinating and open-pollinating genotypes.

## 1. Introduction

Bolivia, Chile, Ecuador, the Galapagos Islands and Peru together constitute the centre of origin and distribution of tomatoes (*S. lycopersicum* L.) [1], although early domestication of tomatoes occurred in both the Andean region and Mexico [2]. Wild tomato relatives are still distributed naturally in the Andean region [3] and may harbour an untapped diversity of genes, which might be useful in diversifying the current cultivated tomato gene pool. In South America, the first domestication, affecting weight and fruit shape, occurred in Ecuador and Peru [4]. In Bolivia, tomato relatives grow in a wide range of ecosystems [5] and public Bolivian institutions are responsible for utilisation, conservation and characterisation of Bolivian core germplasm [6,7]. Some progress has been made in classifying the genetic diversity of Bolivian tomatoes by evaluation and comparison of quality traits among selected accessions. For example, there has been an evaluation and characterisation of 28 tomato accessions from Bolivian core germplasm [8] and a genetic study of 31 accessions from various subspecies of *S. lycopersicum* [9]. A wide range of phenotypic differences has been reported for Bolivian germplasm, despite a lack of evaluations using molecular markers [2,9,10]. However, lack of knowledge on genetic variation in Bolivian tomatoes and their wild relatives impedes the successful use of Bolivian germplasm in breeding programmes.

Recent advances in genomics and molecular technologies have enabled the characterisation of genetic diversity and its effective use in commercial breeding [11]. The genome sequence of the tomato has recently been made available by the Tomato Genome Sequencing Consortium [12,13], greatly facilitating development of genome-based tomato breeding using specific markers. The increasing availability of sequencing data, together with the whole-genome reference sequence, also provides opportunities to develop and use molecular markers for population genetics analyses, including genetic variation within and among tomato populations [14]. Among the molecular markers currently available for population genetics studies, simple sequence repeats (SSR), also referred to as microsatellites, are among the most popular. SSRs are co-dominant markers mainly found outside genes and in non-coding regions of genes [15,16]. These markers have been used to determine polymorphism in tomato landraces in studies with different objectives, such as identifying differences between cultivars [17], conserving representative plant material [18], identifying desirable quantitative traits [19] and promoting pyramidal marker-assisted selection to build disease resistance [20].

Current knowledge on the genetic variation in Bolivian core germplasm is insufficient. Further research is needed to understand the genetic relationship between Bolivian tomato germplasm and the domestication process, and also to unveil desirable quality and agronomic traits [3]. The aim of the present study was thus to determine the genetic diversity of Bolivian tomatoes and their level of relatedness. This was done by analysing 28 accessions of tomato, consisting of four cultivars, four advanced breeding lines and 20 accessions from the Bolivian core germplasm collection. A further aim was to identify possible relationships between the genetic variation of the accessions and variations in their phenotypic traits (fruit size, shape and colour), cultivation status, growth type, geographical region of origin and growing site altitude. 

## 2. Materials and Methods

### 2.1. Plant Material, Planting, Sampling and DNA Extraction

The cultivated and wild tomato germplasm obtained from different sources for this study are referred to hereafter as “accessions” for the sake of simplicity. The 28 tomato accessions analysed represented cultivars, advanced lines, landraces and wild populations (Table 1 and Figure 1). Three hybrid cultivars (‘Lia’, ‘Shanty’, ‘Huichol’) commonly used by tomato farmers in Bolivia were purchased from a local market in Cochabamba, Bolivia. Cultivars ‘Lia’ and ‘Shanty’ are marketed by Hazera Seeds Ltd. and ‘Huichol’ by Seminis. An open-pollinated cultivar (‘Rio Grande’) and four advanced breeding lines were obtained from the Bolivian National Center for Horticultural Seed Production (CNPSH) [16]. The remaining 20 accessions (landraces and wild populations, see Table 1), which were originally collected from the geographical positions shown in Figure 2, were selected from the accessions held at the Horticultural Germplasm Bank—National Germplasm Center—(BGH-BNG) located in CNPSH. The accessions held at the BGH-BNG representing the Bolivian tomato core-germplasm were collected by the personnel involved in the establishment and management of a tomato gene bank. Of the 162 registered accessions at CNPSH collected between 1938 and 2010, only 119 wild relatives or landraces were originally collected in Bolivia (in Beni (7), Cochabamba (44), La Paz (63), Santa Cruz (2), and Sucre (3)) [5]. Of the 44 accessions from the Cochabamba region, only one cultivated accession was included in the present analysis. This is because the wild accessions collected share the exact same geographical location [5]. Pre-selection criteria such as significant geographical distance between accessions combined with region further narrowed the study sample to the most representative 20 accessions from the five regions (Table 1).

Seeds of the 28 accessions were planted in 4.5-L plastic trays filled with nutrient-rich soil (0.08 L per plug) in a greenhouse at the Department of Plant Breeding, Swedish University of Agricultural Sciences (SLU). The emerging seedlings were grown under day/night temperature of ±24 °C/19 °C, 16 h of light and 60% relative humidity until sampling of leaf tissue for DNA extraction at one month after planting. Young leaf tissue was sampled separately from 10 individual plants of all accessions except one, which was represented by nine plants (279 samples in total). The fresh leaf tissue taken from each plant was placed in a separate 2-mL Eppendorf tube containing two glass beads with diameter 3 mm, and immediately dipped in liquid nitrogen. The frozen samples were stored at −80 °C until DNA extraction.

The frozen leaf samples were homogenised for 1 min at 30 Hz in a Mixer Mill (MM400-Retsch GmbH, Haan, Germany). Then 400 μL of CTAB-based buffer (0.1 M Tris-HCl, 20 mM EDTA, 1.4 M NaCl, 2% CTAB, pH 7.5) were added to each sample, followed by incubation for 15 min at 56 °C and centrifugation (using Eppendorf 5427 R, Hamburg, Germany) at 9000× *g* for 3 min. A 200 μL subsample of the supernatant of each sample was transferred to a 96-well plate and DNA extraction was performed with a QIAamp DNA kit, using a QIAcube HT extraction robot (Qiagen, Hilden, Germany). The integrity of the extracted DNA was evaluated by agarose gel electrophoresis (1.5% *w*/*v*) and the quantity was determined using a NanoDrop spectrophotometer (ND-1000, Saveen Werner, Sweden). All extracted DNA samples were kept at −20 °C until polymerase chain reaction (PCR) analysis, while the working solutions (5 ng/μL) were kept at 4 °C for up to 24 h. 

### 2.2. Identification of SSRs in the Tomato Genome and Primer Design 

The genome of the *S. lycopersicum* cultivar ‘Heinz 1706’ (SL3.0 reference Annotation Release 103; (https://www.ncbi.nlm.nih.gov/assembly/GCF_000188115.4 (accessed on 27 July 2022)) was used to identify SSRs in the tomato genome, which were then utilised as new genomic analysis resources for tomato. First, genomic regions of 400 bp to 1200 bp, representing the 12 tomato chromosomes, were randomly sampled. These sequences were searched for identification of dinucleotide and trinucleotide repeat motifs, using WebSat, internet-based software developed for SSR identification [17]. Sequences containing the target SSRs were then further screened based on the suitability of the SSR positions in the sequences for primer design. Next, these sequences were compared against the tomato reference genome at the National Center for Biotechnology Information (NCBI) database, to identify those that are single copy (unique) in the tomato genome, using the Basic Local Alignment Search Tool (BLAST). This resulted in selection of 22 single-copy sequences (two sequences per chromosome) and two highly similar sequences (corresponding to TomSat9-2a and TomSat-2b SSR loci, see Appendix A. The Primer3 program [18,19,20], targeting these SSRs, was used for primer design. 

Ten genotypes were selected from the different tomato accessions for a first testing of the newly designed primer-pairs in amplifying the target SSR loci under optimised PCR conditions (described below). Twelve of the primer-pairs amplified extra fragments in addition to the target loci, and were therefore excluded, while the remaining primer-pairs amplified only their target loci. To confirm that they matched the target sequences, the amplified products of the 12 primer-pairs were purified and sequenced. Thereafter, the PCR products were purified using E.Z.N.A. Cycle Pure Kit V-spin (Omega Bio-tek; Norcross, GA, USA) and 2 μL of 10 μM sequencing primer and 15 μL of 1 ng/μL purified PCR product for each sample were mixed and sent to Eurofins Genomics Sequencing GmbH (Anzinger Str. 7, 85560 Ebersberg, Germany), where Sanger sequencing was conducted. Each amplified product was sequenced with both the forward and reverse primers used for the PCR. The DNA sequences of the PCR products were then aligned with their corresponding reference sequences using CLUSTAL X version [21], which confirmed amplification of the target loci. 

In parallel with the development of the new SSR markers, 40 primer-pairs previously reported to amplify polymorphic SSR loci [22,23,24,25] were screened to determine the quality of their amplified products under optimised PCR reaction conditions. Four of these primer-pairs (a–d in Table 2) were selected for use in this study, together with the 12 newly developed primer-pairs (Appendix A).

### 2.3. PCR Amplification and Electrophoresis

The 16 selected primer-pairs were used to amplify the target loci of 279 individual genotypes from the 28 tomato accessions. The 5′-end of the forward primers was labelled with either 6-FAM or HEX fluorescent dye (Sigma-Aldrich, St. Louis, USA), for detection of amplified products during capillary electrophoresis. A GCTTCT hexamer was added to the 3′-end of the reverse primers (PIG tailing) to prevent the Taq polymerase from adding non-template sequences to the PCR products, as described in Ballard et al. [26]. The PCR reaction solution was prepared for each sample using the following reagents from Thermo Fisher Scientific GmbH (Waltham, MA, USA) (V.A. Graciuno 8. LT-02241 Vilnius, Lithuania): 2.5 μL Dream Taq buffer (KCl, (NH_4_)_2_SO_4_ and 20 mM MgCl_2_), 0.3 μL dNTPs (25 mM), 7.5 μL of each forward and reverse primer (10 μM), 0.2 μL DreamTaq DNA Polymerase (5 U/μL), and 5 μL of 5 ng/μL DNA template. A negative control reaction, replacing the DNA template with sterile Millipore water, was also included. 

The PCR analysis was carried out using a Bio-Rad thermal cycler S1000 (Hercules, CA, USA) with the following cycling parameters: initial denaturation for 3 min at 95 °C, followed by 35 cycles of denaturation for 30 s at 94 °C, annealing for 40 s at 3–5 °C below the primer’s melting temperature and primer extension for 40 s at 72 °C, and a final primer extension for 20 min at 72 °C. After each PCR run, electrophoresis was performed using 1.5% agarose containing GelRed on randomly selected amplified products using a 50 bp Gene Ruler ladder (Thermo Fisher Scientific GmbH, Dreieich, Germany) as size standard, followed by scanning with a BioDoc-It Imaging System (Upland, CA, USA). This led to exclusion of four of the 12 newly developed SSRs, because a significant number of samples failed to be amplified. Hence, the amplified products of 12 SSR loci were used in capillary electrophoresis, as described below. 

The PCR products of the 12 SSR loci were multiplexed into four panels following the criteria described in Geleta et al. [27], except that each PCR product was diluted 1:10 in Millipore ultrapure water before multiplexing. This was followed by mixing each multiplexed PCR product (0.5 μL) with Hi-Di™ Formamide (9 μL) (Thermo Fisher Scientific, Waltham, MA, USA) and Size Standard Gene Scan—600 LIZ (9.7 μL) (Thermo Fisher Scientific, Austin, TX, USA), heating at 96 °C for 3 min and cooling on ice. Multiplexed PCR products were then separated by capillary electrophoresis as described in Andersson et al. [28], using an Applied Biosystems 3500 Genetic Analyzer (Thermo Fisher Scientific, Waltham, MA, USA), at the Department of Plant Breeding, SLU, Sweden.

### 2.4. Data Analysis

The capillary electrophoresis step was followed by peak identification using GeneMarker version 2.7.0 (SoftGenetics, LLC, State College, PA, USA) software with the default settings [29]. Each peak was regarded as an allele and its size was determined using the GS600 size standard. Among the 12 SSR loci, locus TomSatX1-1 was monomorphic across the 28 accessions studied, and hence was excluded from the final data analysis. The alleles of each sample for the remaining 11 polymorphic SSR loci (Table 2) were exported to Excel and converted to genotypic data for subsequent statistical analysis.

Various genetic diversity parameters were estimated for each locus across all accessions and for each accession across all loci. POPGENE version 1.32 [30] was used to determine observed number of alleles (Na), effective number of alleles (Ne) and percentage of polymorphic loci (%PL). GeneAlEx 6.41 [31] was used to determine number of private alleles (NPL), number of locally common alleles (NLCA), expected heterozygosity (He), observed heterozygosity (Ho), Shannon information index (I), genetic differentiation (G_ST_) and fixation index (F). Polymorphism information content (PIC) of each locus was calculated as described in Botstein et al. [32].

Analysis of molecular variance (AMOVA) was conducted to determine the variance within and between accessions and the variance at higher hierarchal level, using Arlequin version 3.5.2.2 [33]. Matrices of pairwise F_ST_ and average pairwise differences between and within accessions were used to generate graphs using a series of R scripts within Rcmd, a console version of the R statistical package, by triggering the command button added to Arlequin version 3.5.2.2 toolbar.

Nei’s unbiased genetic distance between the 28 accessions was calculated using GeneAlEx 6.41 and then used as input data for Unweighted Pair Group Method with Arithmetic Mean (UPGMA)-based cluster analysis using MEGA7 software, where the optimal tree with the sum of branch length −2.13486148 is shown [34,35]. The genetic distance value was also used for principal coordinate analysis (PCoA) using GeneAlEx 6.41 to visualise the genetic relationship between the accessions. Bayesian statistics-based population structure analysis was conducted using STRUCTURE version 2.3.4 [36], based on the admixture model implementing 100,000 burn-in periods and 200,000 Markov chain Monte Carlo chain iterations for K (number of genetic populations) ranging from two to 15 (with 10 independent runs at each K). The optimum K was then determined using STRUCTURESELECTOR [37], a statistical program based on the STRUCTURE output, following the ΔK approach [38]. A β version of CLUMPACK [39] integrated into the STRUCTURESELECTOR was used to visualise the population structure after the optimal K was determined. 

## 3. Results

### 3.1. SSR Markers

The total number of alleles recorded across the 11 SSR loci was 33, with the number of alleles per locus varying from two to five (Table 2). Five of the 11 SSR loci had only two alleles per locus. SLM6-11 and TomSat11-1 were the most polymorphic loci, with five alleles each. The average number of alleles observed per population (Na) for each locus varied from 1.04 (for TomSatX2-2 and TomSatX7-2) to 1.57 (for SLM6-11), with an overall mean of 1.21. The effective number of alleles (Ne) ranged from 1.02 (for TomSatX2-2) to 1.305 (for SLM6-11), with an overall mean of 1.12 (Table 2). The polymorphism information content (PIC) of the loci ranged from 0.05 (for TomSatX2-2) to 0.65 (for SLM6-11), with an overall mean of 0.29. The most informative locus among the 11 loci was SLM6-11 (PIC = 0.65), followed by TomSatX11-1 (PIC = 0.49). Observed heterozygosity (Ho) at locus level varied from zero (for SLM6-11 and TomSatX2-2) to 0.20 (for LE20592), with an average value across the loci of 0.05. Similarly, expected heterozygosity (He) varied from 0.011 (for TomSatX2-2) to 0.178 (for SLM6-11), with a mean of 0.07. The corresponding values for unbiased expected heterozygosity (uHe) for these loci were 0.01 and 0.20, respectively. The SSR loci showed wide variation in their fixation index values, with minimum, maximum and mean values of −0.45, 1.0 and 0.4 for F_IS_, 0.62, 1.0 and 0.87 for F_IT_, and 0.72, 0.90 and 0.80 for F_ST_. Estimated genetic differentiation (G_ST_) at locus level varied from 0.67 (for SLR20) to 0.89 (for TomSatX11-1), with a mean of 0.77, and was highly significant at all loci (*p* = 0.001) (Table 2).

**Table 2 genes-13-01505-t002:** Total number of alleles (TNA), number of different alleles (Na), effective number of alleles (Ne), polymorphism information content (PIC), observed heterozygosity (Ho), expected heterozygosity (He), unbiased expected heterozygosity (uHe), fixation indices (F_IS_, F_IT_ and F_ST_), and population differentiation (G_ST_, an analogue of F_ST_ adjusted for bias) and its *p*-value (P(G_ST_)), for each SSR locus.

Locus	TNA	Na	Ne	PIC	Ho	He	uHe	F_IS_	F_IT_	F_ST_	G_ST_	P(G_ST_)
SSR22 ^a^	3	1.21	1.18	0.27	0.117	0.080	0.087	−0.45	0.62	0.74	0.72	0.001
SLR20 ^b^	3	1.18	1.09	0.19	0.020	0.056	0.063	0.65	0.90	0.72	0.67	0.001
SLM6-11 ^c^	5	1.57	1.31	0.65	0.000	0.178	0.198	1.00	1.00	0.75	0.69	0.001
LE20592 ^d^	4	1.43	1.28	0.51	0.198	0.150	0.165	−0.32	0.65	0.73	0.71	0.001
TomSatX2-2	2	1.04	1.02	0.05	0.000	0.011	0.013	1.00	1.00	0.79	0.75	0.001
TomSatX7-1	3	1.18	1.12	0.30	0.080	0.069	0.077	−0.16	0.76	0.79	0.77	0.001
TomSatX7-2	2	1.04	1.04	0.10	0.012	0.018	0.021	0.33	0.88	0.82	0.80	0.001
TomSatX8-1	2	1.29	1.14	0.33	0.042	0.090	0.102	0.53	0.90	0.79	0.75	0.001
TomSatX9-2a	2	1.07	1.02	0.13	0.007	0.015	0.017	0.53	0.95	0.89	0.87	0.001
TomSatX9-2b	2	1.07	1.02	0.13	0.007	0.015	0.017	0.53	0.95	0.89	0.87	0.001
TomSatX11-1	5	1.18	1.10	0.49	0.013	0.049	0.055	0.75	0.98	0.91	0.89	0.001
Mean		1.21	1.12	0.29	0.045	0.067	0.074	0.40	0.87	0.80	0.77	0.001
SE		0.025	0.017	0.19	0.010	0.009	0.010	0.15	0.04	0.02	0.03	

^a^ Frary et al. [23], ^b^ Korir et al. [25], ^c^ Geethanjali et al. [24], ^d^ Smulders et al. [22].

### 3.2. Genetic Diversity of the Accessions

The genetic diversity of each accession was estimated using several parameters (Table 3). Accessions BOL-8223-HT, BOL-8281-HT, BOL-8284-HT, BOL-8328-HT, BOL-8330-HT and BOL-8340-HT were homozygous for a single allele at each of the 11 loci. Hence, they had observed Na and Ne values of one, and percentage of polymorphic loci (PPL), Shannon information index (I), Ho, He and uHe values of zero. The highest value of Na (1.64) and Ne (1.38) was recorded for accession ‘HT-25’ and BOL-8288-HT, respectively. Analysis of the number of private alleles (NPA) revealed that accessions BOL-8222-HT, BOL-8225-HT, BOL-8282-HT, BOL-8335-HT and BOL-8348-HT have a single private allele (NPA = 0.09) at locus SSR22, SLM6-11, LE20592, TomSatX2-2 and TomSatX11-1, respectively (Table 3). Of the 28 accessions, 79% had alleles shared by ≤25% of the accessions (NLCA ≤ 25%), whereas all accessions had alleles shared by ≤50% of the accessions (NLCA ≤ 50%). The highest NLCA ≤ 25% value (0.46) was recorded for accession BOL-8288-HT, while the highest NLCA ≤ 50% value (0.64) was recorded for two accessions (‘HT-25’ and BOL-8288-HT) (Table 3).

Among the 28 accessions, six did not have polymorphic loci (PPL = 0), as indicated above, whereas less than 10% of the loci were polymorphic in six other accessions (PPL < 0.1) (Table 3). The advanced breeding line ‘HT-25’ was the only accession with more than 50% polymorphic loci (PPL = 0.55). The second highest PPL value (0.45) was recorded for accessions BOL-8288-HT and BOL-8348-HT, both of which are wild. The mean PPL value for the accessions was 0.19, indicating that only 19% of the loci were polymorphic on average.

The genetic diversity of each accession was estimated by Shannon’s I and He (gene diversity). In addition to the six accessions that did not show genetic variation (see above), 14 accessions had very low genetic variation, with I and He values below 0.14 and 0.10, respectively (Table 3). Four accessions, ‘HT-25’ (advanced breeding line), BOL-8226-HT (cultivated), BOL-8348-HT (wild) and BOL-8288-HT (wild), had relatively high genetic variation, with I and He values above 0.20 and 0.14, respectively (Table 3). For example, accession BOL-8288-HT (wild) had I, He and uHe values of 0.29, 0.21 and 0.25, respectively. The mean values of I, He and uHe for the 28 accessions were 0.10, 0.07 and 0.07, respectively. The vast majority of the accessions (86%) had observed heterozygosity (Ho) below 0.10, including those that were totally homozygous (Ho = 0). ‘Shanty’ (a commercial cultivar from Israel) was the most heterozygous accession, followed by Bolivian advanced breeding line ‘HT-25’, ‘Lia’ (a commercial cultivar from Israel) and ‘Huichol’ (a commercial cultivar from Thailand), with Ho values of 0.27, 0.24, 0.18 and 0.18, respectively (Table 3). The fixation index (F) of the accessions that have polymorphic loci varied from −1.00 (the lowest possible value) in accession ‘Lia’, Shanty and ‘HT-36’ to 1.0 (the highest possible value) in six Bolivian accessions, including one advanced breeding line, one wild and four cultivated accessions (Table 3). 

### 3.3. Analysis of Molecular Variance (AMOVA) 

AMOVA was performed using 1000 permutations at both the accession and higher hierarchical levels (Table 4). The results revealed that 77.3% of the total variation was attributable to variation between accessions (F_ST_ = 0.773, *p* < 0.001), while 22.7% was attributable to variation within accessions, of which 7.1% was accounted for by variation among individuals within accessions and 15.6% by variation within individuals. 

The AMOVA analysis at higher hierarchical level was performed by grouping the accessions using seven different criteria: geographical region of origin (La Paz and other regions (Cochabamba, Sucre, Santa Cruz and Beni)); altitude (<500 m above sea level (masl), 950–1200 masl, 1450–1750 masl, and 1850–-2250 masl); cultivation status (cultivated and wild); fruit shape (cylindrical, round, slightly flattened); fruit colour (red and yellow); fruit size (intermediate and very small); and growth type (determinate, semi-determinate and indeterminate). Accessions from La Paz differed significantly from those of the other regions in Bolivia, and region of origin accounted for 21.7% of the total variation revealed by the markers used (*p* < 0.001). Cultivated and wild accessions also differed significantly, with cultivation status accounting for 17.9% of the total variation (*p* < 0.001) (Table 4). In addition, there were significant genetic differentiations into fruit shape groups, fruit size groups and growth type groups, accounting for 16.8%, 22.5% and 22.0% of the total variation, respectively. However, no significant differences were recorded for altitude groups and fruit colour groups (Table 4). 

Pairwise F_ST_ analysis of the 28 accessions revealed significant differentiation (*p* < 0.05) among the vast majority (94%) of the pairs of accessions (Figure 3). The three non-Bolivian accessions (‘Lia’, ‘Shanty’ and ‘Huichol’) showed significant differentiation from the Bolivian accessions. The differentiation among the four Bolivian advanced breeding lines was also significant (*p* < 0.05) (Figure 3). ‘Rio Grande’, a widely cultivated variety in Bolivia, showed significant differentiation from all other accessions except ‘HT-23’ and BOL-8224-HT. ‘Rio Grande’ and BOL-8224-HT showed no significant differentiation from each other, having an F_ST_ value close to zero (marked with a purple asterisk in Figure 3). Other pairs of accessions that were not significantly differentiated included BOL-8328-HT versus BOL-8330-HT (semi-determinate wild accessions from La Paz with very small red fruits), and BOL-8316-HT (a cultivated accession from La Paz) versus ‘HT-36’ and ‘HT-37’ (advanced breeding lines) (Figure 3). Accessions BOL-8281-HT, BOL-8335-HT, and BOL-8349-HT (wild accessions from Beni, La Paz and La Paz, respectively) showed high differentiation from most accessions studied, as revealed by pairwise F_ST_ (Figure 3). 

A heatmap of pairwise Nei’s distance (below diagonal in Figure 4) corroborated the low differentiation between most accessions except BOL-8281-HT, BOL-8284-HT, BOL-8335-HT and BOL-8349-HT. These accessions had higher genetic distance to most other accessions, as can also be observed from the heatmap of pairwise F_ST_ (Figure 3). In line with the Shannon’s I and He values (Table 3), the within-accession variation was zero for accessions BOL-8223-HT, BOL-8281-HT, BOL-8284-HT, BOL-8328-HT, BOL-8330-HT and BOL-8340-HT (Figure 4). In addition, extremely low variation was recorded within accession BOL-8349-HT. Accessions ‘HT-25’, BOL-8288-HT and BOL-8348-HT were the most diverse (Figure 4), again in agreement with the Shannon’s I and He values.

### 3.4. Cluster Analysis and Principal Coordinate Analysis

The UPGMA cluster analysis based on Nei’s unbiased genetic distance revealed various genetic relationships among the 28 tomato accessions, including four clusters and two solitary accessions (Figure 5). Cluster 1 comprised five wild accessions with semi-determinate growth habit from La Paz (BOL-8328-HT, BOL-8330-HT, BOL-8348-HT, BOL-8349-HT and BOL-8335-HT) and one cultivated accession with indeterminate growth habit from Beni (BOL-8284-HT). Cluster 2 comprised two phenotypically similar wild accessions (BOL-8288-HT from La Paz and BOLTH-0119-HT from Beni). A wild accession from Beni (BOL-8282-HT) with indeterminate growth habit and a cultivated accession from La Paz (BOL-8340-HT) with semi-determinate growth habit remained solitary, forming clusters 3 and 4, respectively. Cluster 5 comprised five cultivated accessions, four of which have a determinate growth habit (BOL-8222-HT, ‘HT-23’ and ‘Rio Grande’ from Cochabamba and BOL-8224-HT from Santa Cruz), and one semi-determinate type from Sucre (BOL-8223-HT). Cluster 6 was the largest cluster, comprising 13 accessions that were further divided into three sub-clusters. The first sub-cluster comprised three semi-determinate accessions, two of which were from Sucre (BOL-8225-HT and BOL-8226-HT; both cultivated) and one from La Paz (BOL-8292-HT; wild). The second sub-cluster comprised four accessions with determinate growth habit that included a cultivated accession from La Paz (BOL-8295-HT) and the three foreign commercial cultivars (‘Lia’, ‘Shanty’ and ‘Huichol’). The third sub-cluster comprised three advanced breeding lines with determinate growth habit from Cochabamba (‘HT-25’, ‘HT-36’ and ‘HT-37’) and three semi-determinate accessions from La Paz (two cultivated: BOL-8316-HT and BOL-8322-HT, and one wild: BOL-8290-HT). 

The Nei’s unbiased genetic distance-based principal coordinate analysis (PCoA) further revealed the genetic relationship between the 28 tomato accessions (Figure 6). The first two principal coordinates (PCoA) together explained 70% of the total variation, with PCoA1 explaining 52.3% and PCoA2 17.2%. The six accessions in cluster 1 of the UPGMA tree (Figure 5) formed two small clusters (highlighted in sky blue and green in Figure 6). The two accessions in cluster 2 of the UPGMA tree (BOL-8281-HT and BOL-8288-HT) were separated along PCoA2. Three of the five accessions in cluster 5 of the UPGMA tree (‘HT-23’, ‘Rio Grande’ and BOL-8224-HT) were separated along PCoA1, forming a group highlighted in pink in Figure 6. The remaining two accessions in cluster 5 (BOL-8222-HT and BOL-8223-HT), the solitary accessions BOL-8282-HT and BOL-8340-HT, and all accessions in cluster 6 of the UPGMA tree formed the group highlighted in yellow in Figure 6. 

### 3.5. Population Structure Analysis

Analysis of admixture model-based population structure using the STRUCTURE and STRUCTURESELECTOR programs revealed that three genetic clusters (K) was the optimal number, according to the method of Evanno et al. [38] (Appendix A). This suggests that the 279 individuals from the 28 accessions analysed in this study originated from three genetic populations. The graphical illustration of the population structure of the 28 accessions clearly showed that most were admixed, albeit to varying degrees. BOL-8335-HT and BOL-8249-HT were the only accessions that were not admixed (Figure 7). All accessions that formed cluster 1 in the UPGMA tree (Figure 5) were represented by deep-purple-dominated bars except BOL-8348-HT, which appeared to show high admixture. Similarly, all accessions that formed cluster 6 in the UPGMA tree were represented by blue-dominated bars, except accessions BOL-8225-HT and BOL-8226-HT (Figure 7). Other highly admixed accessions were BOL-8225-HT, BOL-8282-HT and BOL-8288-HT. Among highly admixed accessions, BOL-8282-HT was the only one with significant proportions of alleles from the three genetic clusters represented by the three different colours in Figure 7. In general, the results of cluster, PCoA and population structure analyses were in good agreement regarding the genetic relationships between the accessions studied.

## 4. Discussion

Our analysis of 28 tomato accessions revealed a low level of diversity persisting in exanimated Bolivian tomatoes, despite the fact that Bolivia is part of the centre of diversity [40] or origin [1,41] of tomatoes. The diversity identified in the analysis was mainly between accessions, with factors such as geographical region of origin, cultivation status, fruit shape, fruit size and growth type clearly dividing the tomatoes into groups that were genotypically differentiable by the markers used. Thus, the markers developed within this study and their relationship to the various parameters considered provide opportunities to select parents for crossbreeding in tomato breeding programmes, which is an important measure to generate new Bolivian cultivars [42]. The present study also contributed novel knowledge on the genetic diversity and population structure of Bolivian tomatoes, as previous studies have only included Bolivian accessions of cherry tomatoes [43] and its wild relatives *S. lycopersicum* var. *ceraciforme* and *S. neorickii* [9]. This novel knowledge increases understanding of genetic relationships among Bolivian tomato germplasm and the domestication processes of tomatoes.

### 4.1. The SSR Markers in Revealing Tomato Genetic Diversity

Use of SSR loci previously employed to study the genetic diversity of tomatoes revealed a considerably lower number of alleles for the Bolivian tomatoes investigated here than reported for tomatoes from other countries. For instance, only three alleles were detected at the SLR20 locus among the 28 tomato varieties studied here, whereas Korir et al. (2014) reported five alleles among 42 tomato varieties from China and Kenya at this locus [25]. Further, Gonias et al. (2019) reported eight alleles at this locus in their study of 107 tomato accessions, including cultivars and landraces from Greece and international hybrids [44]. At the SLM6-11 locus, five alleles were observed in this study, whereas six alleles were reported by Geethanjali et al. (2010) for 16 tomato accessions [24]. Smulders et al. (1997) identified seven alleles at the LE20592 locus for 10 tomato accessions encompassing seven *S. lycopersicum* cultivars and three wild *Solanum* species [22], but in the present study only four alleles were recorded at this locus. Some of these previous studies analysed a higher number of accessions than in the present study, while other studies analysed fewer accessions, but a higher number of alleles was reported in all cases. In light of this, Bolivian tomatoes can be considered to have low allelic diversity.

Among the 11 SSR loci studied, SLM6-11 was the most informative, with a PIC value of 0.65, corresponding to the value reported in a previous study [24]. Hence, for population genetics analysis of tomato genetic resources, this locus should be prioritised, along with LE20592 (PIC = 0.55) and TomSatX11-1 (PIC = 0.49), the latter developed in the present study. As in a previous study [45], which reported PIC values ranging from 0.06 to 0.60 (mean 0.31) for inbred tomato lines from different countries, the PIC value in this study ranged from 0.05 to 0.65, with a mean of 0.29. However, higher PIC value ranges and mean values have been reported in other studies, e.g., 0.62–0.85 (mean 0.74) for diverse tomato varieties of modern, landrace and hybrid type using SSR markers [44], 0.42–0.87 (mean 0.69) for different tomato species using SSR markers and 0.17–0.74 (mean) 0.45 for tomato varieties from different countries [25,46]. Despite the fact that the PIC values are directly related to the choice of SSRs, the results of the present study indicate that the genetic diversity of Bolivian tomatoes is relatively low. However, the genetic differentiation between the accessions was highly significant, as shown by the high F_ST_ and G_ST_ values at each SSR locus. Highly significant differentiation between tomato accessions has been reported previously for local landraces from southern Italy and contemporary tomato varieties [47], and for tomato landraces from Cyprus, France and Greece [48].

Tomato is predominantly a self-pollinating species [49,50], and for such species observed heterozygosity (Ho) should normally be lower than expected heterozygosity (He) at a neutral polymorphic locus. In line with expectations, the He values were higher than the Ho values for most of the SSR loci analysed in this study. However, Ho exceeded He for the SSR22, LE20592 and TomSatX7-1 loci, indicating that these loci might be linked to genes under balancing selection, which favours heterozygosity [51]. The He values for the different accessions in the present study were generally low (mean 0.07), whereas higher values (0.17–0.71) have been reported in previous studies [2,52,53]. The significant number of open- and self-pollinated accessions included in the present study might be the reason for the low average He values found, as a higher level of heterozygosity can be expected for hybrids [54] than for open- or self-pollinated accessions [55]. Based on available information, only seven (‘Lia’, ‘Shanty’, ‘Huichol’, and the four advanced breeding lines) of the 28 accessions were hybrids, while the others were open- or self-pollinated. Correspondingly, the commercial hybrid cultivars ‘Lia’, ‘Shanty’, and ‘Huichol’ had higher Ho values (0.27, 0.18, and 0.18, respectively) than the other accessions (Ho < 0.1), while the open/self-pollinated group had Ho values that ranged from zero to 0.05. As expected, Ho was lower or equal to He for the open/self-pollinated group.

For the hybrid group, Ho exceeded He except for ‘HT-23’ and ‘HT-37’, with both having higher He than Ho and high positive fixation index (F). Steps taken following hybridisation could explain the differences seen in F and He/Ho ratio. One possibility is that ‘HT-23’ and ‘HT-37’ have undergone a series of self-pollination events after the hybridisation event, resulting in homozygosity at the majority of their loci and positive F values. In contrast, the two Israeli commercial cultivars (‘Lia’ and ‘Shanty’) and the Bolivian hybrid ‘HT-36’ are most likely F1 hybrids, since they possess maximum levels of heterozygosity (F = −1), while ‘Huichol’, a cultivar from Thailand, and the Bolivian hybrid ‘HT-25’ may have been reproduced through open pollination after the hybridisation event took place. The 20 Bolivian accessions can be classified into three subgroups based on their expected heterozygosity, which measures the genetic diversity within the accessions: those with He = 0 (inbred), those with He = 0.01–0.08 (extremely low diversity), and those with He = 0.11 = 0.21 (low to medium diversity). Shannon’s I values can also be used to discern these subgroups. However, in terms of phenotypic characteristics and geographical region of origin, each of these groups was found to be generally diverse. For example, the six inbred accessions differ in fruit colour and shape, as well as altitude and geographical region of origin of the germplasm. That group also comprised both cultivated and wild types. Additionally, some of those with very small fruits appeared to be inbred, while others had He values as high as 0.21. Hence, neither geography nor phenotypic characteristics sufficiently explained the genetic variation within the accessions.

The results of the present analysis indicated differences in reproductive mechanisms among both cultivated and wild Bolivian tomatoes. The accessions BOL-8223-HT, BOL-8281-HT, BOL-8284-HT, BOL-8328-HT, BOL-8330-HT and BOL-8340-HT are genetically uniform, and are characterised by very small red fruits except for BOL-8223-HT (which has very small yellow fruits). Their lack of within-accession genetic variation may indicate that they have cleistogamous flowers that prevent pollen movement, resulting in strict inbreeding. Crossbreeding such inbred accessions can be advantageous, since they will produce genetically suitable F1 hybrids that may be superior to their parents in terms of desirable traits. It is possible, for example, to crossbreed BOL-8223-HT (cultivated) with BOL-8330-HT (wild), since they are genetically distinct, as revealed by our cluster, PCoA and pairwise F_ST_ analyses, while they show some differences in their phenotypic characteristics.

The accessions BOL-8225-HT, BOL-8282-HT, BOL-8290-HT, BOL-8316-HT, BOL-8322-HT) and ‘HT-23’ have similar characteristics to the above group except that they have genetic variation within accessions. However, the high fixation index values obtained here (F = 1) indicate that these accessions are strictly self-pollinating types, a reproductive mechanism determined in previous studies to be dominant in tomatoes [2,49]. In view of the fact that they are cultivated types with the exception of BOL-8290-HT, the low level of genetic variation within these accessions might be due to unintentional gene flow in the form of seeds. The accessions with high He and PPL values, i.e., BOL-8226-HT, BOL-8288-HT and BOL-8348-HT, are most likely open-pollinating types with a high rate of outcrossing. 

The two most genetically diverse Bolivian accessions, BOL-8288-HT (He = 0.21; PPL = 0.45) and BOL-8348-HT (He = 0.19; PPL = 0.45), are both wild accessions bearing small round fruits. They differ in fruit colour (red and yellow, respectively) and flowering habit (indeterminate and semi-determinate, respectively) and in the altitude and geographical location of the sampling site. Further characterisation may lead to identification of genotypes with desirable characteristics that can be incorporated into elite cultivars through crossbreeding. The present study clearly indicated that none of the specific geographical regions or altitude ranges within Bolivia can be considered a hotspot for the genetic diversity of Bolivian tomatoes.

### 4.2. AMOVA 

The analysis of molecular variance (AMOVA) results for the 28 *S. lycopersicum* accessions revealed significant variation both between accessions (77.3%) and within accessions (22.7%), which is in line with the characteristics of species that are predominantly self-pollinating. A previous study [56] that evaluated two wild tomato species from the Galapagos Islands (*S. cheesmaniae* and *S. galapagense*) also revealed much higher variation between accessions (>90% of the total genetic variation) than within accessions. In fact, these two wild species are strict inbreeders, unlike some open-pollinating *S. lycopersicum* accessions analysed in the present study. However, higher variation within accessions (accounting for 29% and 36% of the total variation, respectively) has previously been reported for *S. lycopersicum* and *S. pimpinellifolium*, both being self-compatible [50]. In contrast, in the outcrossing *S. peruvianum*, only 32.2% of the total variation is between accessions [50]. A possible conclusion from the above discussion is that the reproductive mechanism of a species can have a profound impact on population differentiation. 

Based on the highly significant genetic variation found between the accessions and the significant differentiation between over 90% of accession pairs, crossbreeding between genotypes bearing desirable agronomic and fruit characteristics may prove to be the most effective approach for cultivar development. It should be noted that in the present study, hierarchical AMOVA revealed significant differences between groups based on a variety of factors, such as region of origin, domestication/breeding status, fruit shape, fruit size and flowering habit. Around 20% of the total genetic variation differentiated accession groups from La Paz versus other regions in Bolivia, pointing to the importance of isolation by distance and geographical barriers in population differentiation. This significant divergence can also be explained partly by the fact that most of the accessions from La Paz are wild populations with small fruits and semi-determinate flowers, while most of the accessions from other regions are cultivated types. This differentiation level is comparable with that of *S. chesmaniae*, but four times higher than that of *S. galapagense* from different regions of the Galapagos Islands [56]. On the other hand, the lack of genetic differentiation among altitude groups despite the wide range of altitude of their sampling sites (227–2858 masl) indicates that tomatoes can adapt to altitude. The AMOVA analysis did not significantly differentiate tomatoes with red and yellow skin colour.

Although cultivated Bolivian tomatoes differed significantly from their wild counterpart, that differentiation explained only 18% of the total variation. The majority of the cultivated tomatoes still bear small fruits, similar to wild types, indicating that little has been accomplished in terms of selection-based improvement in fruit size. Significant variations were observed between fruit shape groups, fruit size groups and flowering habit groups, accounting for 16.8%, 22.5% and 22.0% of the total variation, respectively. These results are not surprising, since the commercial cultivars and advanced breeding lines have predominantly cylindrical and round fruits, while the majority of the wild accessions have somewhat flattened fruits. Additionally, commercial cultivars and advanced breeding lines have larger fruits and determinate flowers, while wild and landrace accessions have small fruits and indeterminate/semi-determinate flowers.

Since determinate growth habit and larger fruits are desired characteristics in tomato cultivars, resulting in synchronous maturity that facilitates harvesting and higher fruit yields, these traits have been the target of domestication and breeding programmes [57,58]. Therefore, the significant differences observed here between flowering habit groups and fruit size groups relate to domestication status. The lack of significant differentiation between the fruit colour groups can be explained by the fact that both cultivated and wild accessions possess a large proportion of red fruits, but yellow fruits are also found in both cultivated and wild groups. Tomato skin colour is a phenotypic quality trait that is known to be regulated by several genes, including phytoene synthase 1 (PSY1), phytoene desaturase (PDS), 15-cis-zeta-carotene isomerase (ZISO) and DE-ETIOLATED 1 (DET1) [59]. Due to the fact that the SSRs used in this study are not linked to genes that regulate fruit colour, which is needed to differentiate this trait [60], the results indicate that cultivated tomatoes with different fruit colours have rather similar genetic backgrounds, as is also likely to be the case for wild tomatoes. 

### 4.3. Cluster, Principal Coordinate and Population Structure Analyses

There was significant genetic differentiation between cultivated and wild accessions, with the exception of BOL-8282-HT and BOL-8284-HT, as visualised by the heatmaps of pairwise comparisons, UPGMA tree, PCoA scatter plot and STRUCTURE graph. Thus, use of wild accessions in crossbreeding with cultivated accessions of tomatoes provides the potential to develop cultivars that have favourable fruit characteristics and are well-adapted to Bolivian agroecosystems. It should be noted, however, that there are higher genetic similarities between the Bolivian cultivated tomatoes and the foreign commercial cultivars evaluated here than between Bolivian cultivated and wild tomatoes. This is an excellent example of how domestication has shaped crop evolution through the selection of germplasm for multiple desirable traits that are generally categorised as “domestication syndrome” traits [61]. Furthermore, the determinate and indeterminate types of tomatoes were found to be clearly separated from each other regardless of their cultivation status, unlike the semi-determinate types. Thus, indeterminate cultivated accessions clustered with indeterminate wild accessions, whereas determinate wild accessions clustered with determinate cultivated accessions. This suggests that the substantial genetic differentiation between determinate and indeterminate varieties predates tomato domestication.

Among the cultivated accessions, BOL-8284-HT was the most genetically distinct, exhibiting an indeterminate flowering habit and very small, slightly flattened red fruits. BOL-8335-HT and BOL-8349-HT were the most genetically distinct wild accessions, both exhibiting a semi-determinate flowering habit and producing very small red fruits. Consequently, these accessions could be valuable for crossbreeding to facilitate the development of superior genotypes through genetic recombination for further breeding.

It is noteworthy that some closely clustered accessions, e.g., BOL-8328-HT vs. BOL-8330-HT, ‘Rio Grande’ vs. ‘HT-23’ and ‘HT-36’ vs. ‘HT-37’, have similar fruit characteristics and flowering habits, suggesting the presence of genetically similar accessions that could be considered “duplicates” in the Bolivian ex situ conserved tomato germplasm. In contrast, some closely clustered accessions (high genetic similarity) such as BOL-8316-HT and ‘HT-37’ exhibited different fruit characteristics and flowering habit. Thus, grouping the accessions based on their phenotypic characteristics, followed by genotypic characterisation based upon single nucleotide polymorphisms (SNPs) and SSRs, would allow the creation of a core collection comprising distinct accessions.

In previous population genetics studies, different approaches have been used to determine how populations are genetically structured [36,62,63,64]. The Bayesian model-based population structure analysis assumes that populations are defined by the frequencies of alleles at multiple loci [36]. Using this method, each genotype within a predefined population is assigned to a cluster or, if the genotype is found to be admixed, to more than one cluster. Using this approach, we found that the majority of the 28 accessions studied and their individual genotypes exhibited varying degrees of genetic admixture, suggesting significant gene flow between the different groups. Such genetic admixture may result from natural gene flow in wild habitats and agroecosystems and from intentional crossbreeding to produce cultivars that meet desired characteristics such as fruit quality [57].

The commercial cultivars and most of the Bolivian advanced breeding lines displayed highly similar population structures to wild accessions collected from La Paz region, implying that the core germplasm collected in La Paz plays a significant role in tomato improvement programmes in Bolivia. To our knowledge, only one previous study has examined the genetic diversity of Bolivian core germplasm, using 31 accessions of *S. neorickii*, *S. chmielews*, *S. lycopersicum* ceraciforme and S. *lycopersicum* spp. [9]. That study reported genetic distance in collections between La Paz and other regions. In the present study, not all wild and landraces accessions from Bolivian core germplasm were used, as only accessions classified as *S lycopersicum* L. were included. The best-represented region was La Paz, which had a significant number of accessions in the core collection. Nonetheless, this study provides valuable novel information on genetic diversity and genetic structure for potential use in breeding programmes. Most of the alleles found in the cultivated accessions derived from two of the three genetic populations identified. Alleles of these genetic populations appear to be widely distributed in non-Bolivian and Bolivian accessions, since they are well represented in these accessions. The accessions that formed cluster 1 in the UPGMA tree, shown as sky blue and green clusters in the PCoA plot in Figure 6 (clearly differentiated along PCoA1) were represented by deep purple bars in the optimal genetic structure plot in Figure 7, except for BOL-8348-HT which apparently had high genetic admixture. The results presented here are partly consistent with those of another study in which tomato wild relatives and landraces showed less admixture than market-oriented cultivars [65]. As a whole, the results of the cluster, PCoA and population structure analyses agreed very well, clearly demonstrating genetic relationships between the accessions studied and the pattern of their genetic variation. 

## 5. Conclusions

The number of alleles detected in this SSR-based study on Bolivian tomato accessions ranged from two to five, indicating low allelic diversity of examined Bolivian accessions. TomSatX11-1 proved to be the most informative of the newly developed SSR markers in this study and should be prioritised for population genetics analysis of tomatoes, together with other highly informative markers such as SLM6-11 and LE20592. While Bolivia lies within the centre of diversity and origin of tomatoes, explored Bolivian tomatoes have generally low genetic variation within each accession. However, there is highly significant genetic differentiation between the accessions, explaining approximately 75% of the total genetic variation. There is also significant genetic variation between wild and cultivated tomatoes and between tomatoes with different geographical origin, fruit shape, fruit size and flowering habits. However, there is no significant genetic difference between tomatoes from different altitude ranges or tomatoes with different fruit colours. The genetic differentiation between tomatoes with determinate and indeterminate flowers may predate tomato domestication. Tomatoes have genetically determined mechanisms that contribute to either cleistogamous flowers, generating genetically uniform genotypes, here represented by six accessions (BOL-8223-HT, BOL-8281-HT, BOL-8284-HT, BOL-8328-HT, BOL-8330-HT and BOL-8340-HT), or to flowers that allow open pollination, here represented by three accessions (BOL-8226-HT, BOL-8288-HT and BOL-8348-HT). The two most genetically diverse Bolivian accessions in the present study were BOL-8288-HT (He = 0.21; PPL = 0.45) and BOL-8348-HT (He = 0.19; PPL = 0.45). There is limited gene flow both within and between cultivated types and wild populations, resulting in genetic admixture. Crossbreeding of genotypes of genetically distinct cultivated accessions, such as BOL-8284-HT and BOL-8316-HT, could lead to the development of superior cultivars through genetic recombination.

## Figures and Tables

**Figure 1 genes-13-01505-f001:**
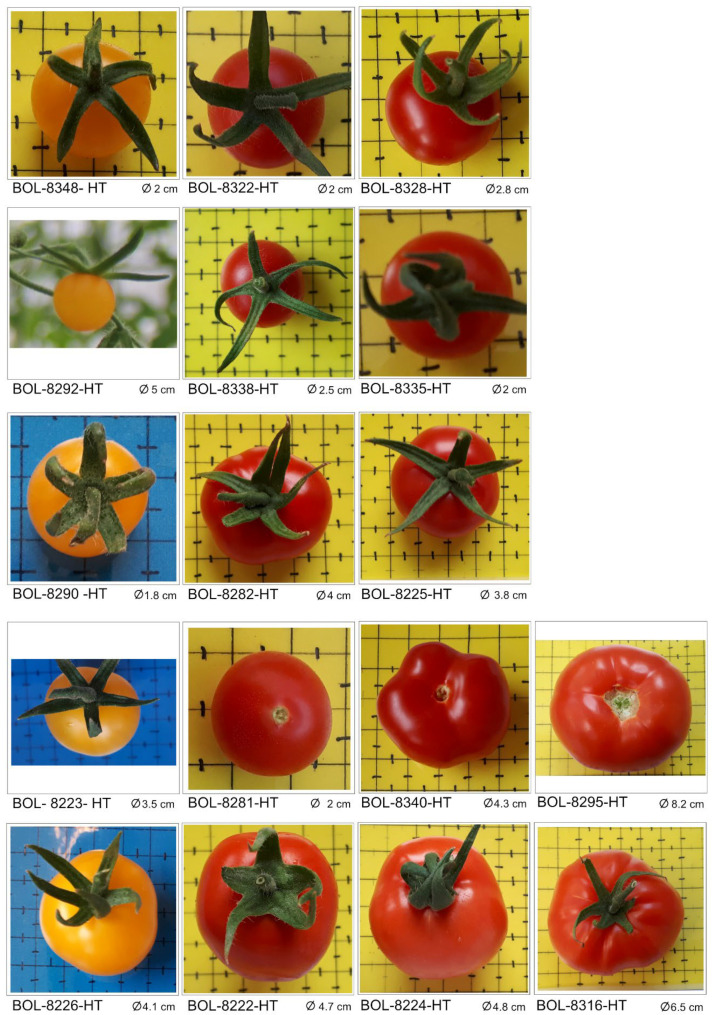
Images of 17 mature tomato fruits from Bolivian core germplasm.

**Figure 2 genes-13-01505-f002:**
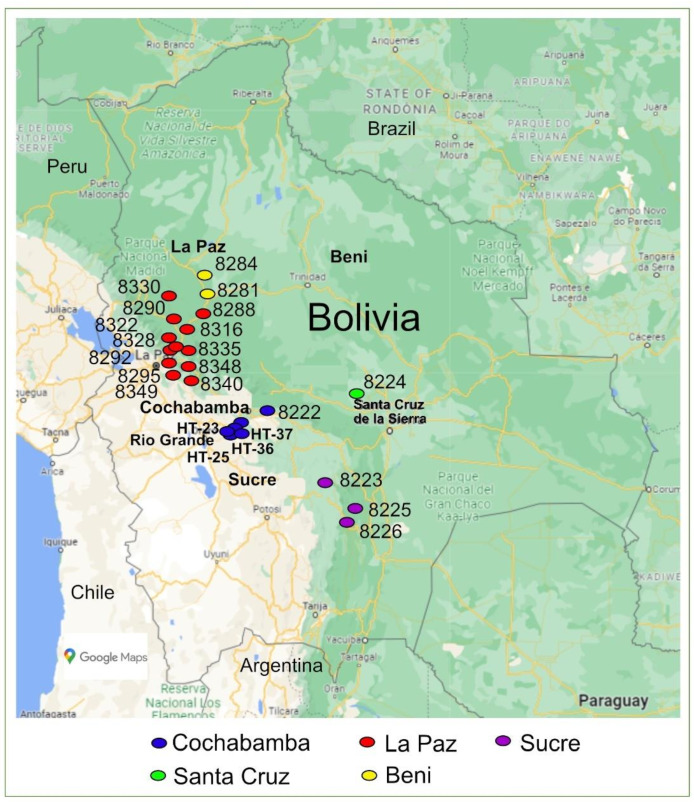
Geographical source of the 25 accessions collected or developed in Bolivia.

**Figure 3 genes-13-01505-f003:**
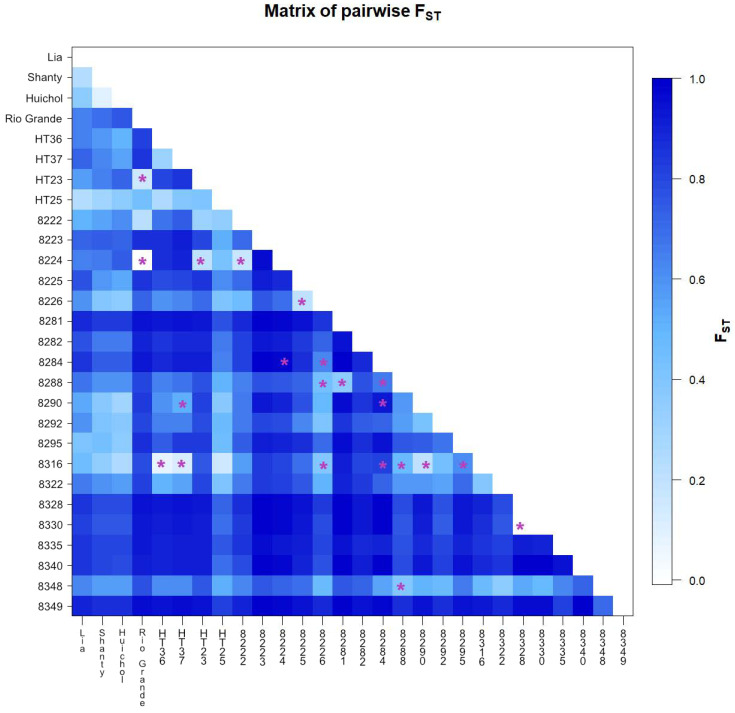
Heatmap of pairwise fixation index F_ST_ of the 28 tomato accessions, calculated using the number of different alleles as a distance method. The differentiation between each pair of accessions was significant (*p* < 0.05) except in the case of pairs marked with a purple asterisk.

**Figure 4 genes-13-01505-f004:**
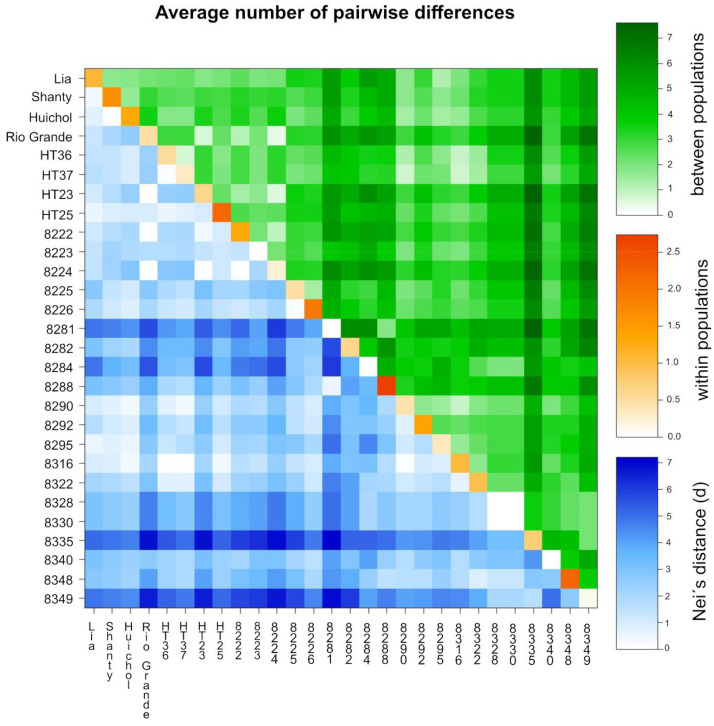
Heatmap displaying average number of pairwise differences of the 28 accessions, estimated using a number of different alleles as a distance method: average number of pairwise differences between the accessions (PiXY; above diagonal), average number of pairwise differences within the corresponding accession (PiX; diagonal); and corrected average pairwise difference (PiXY − (PiX + PiY)/2; below diagonal), also referred to as Nei’s distance (d).

**Figure 5 genes-13-01505-f005:**
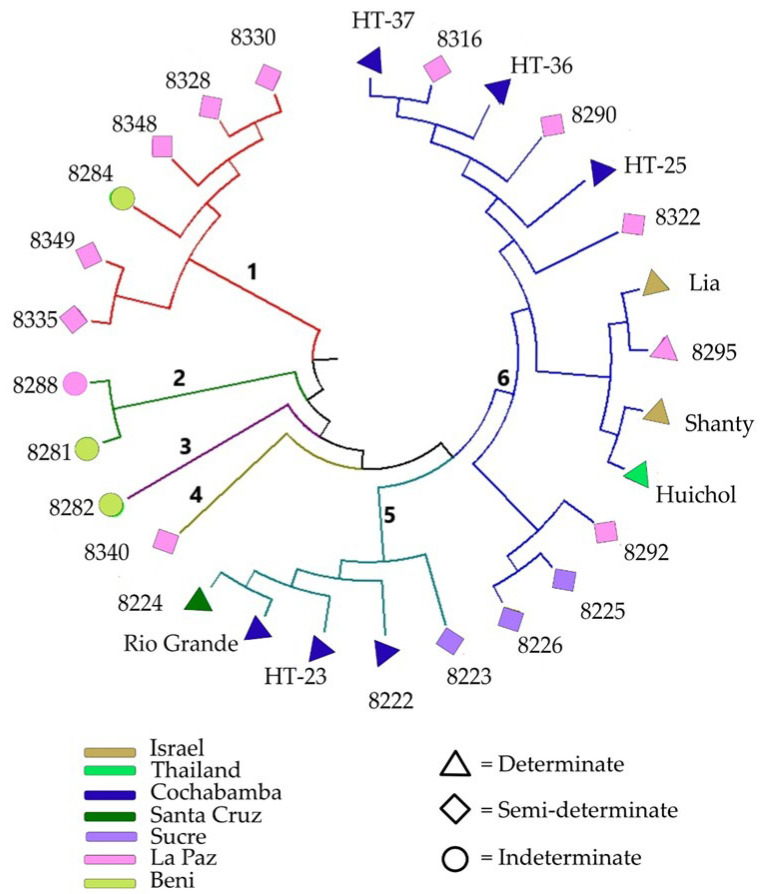
Unweighted pair group method with arithmetic mean (UPGMA) tree showing the genetic relationship between the 28 accessions analysed in the present study. NOTE: Label colour indicates geographical origin of the accessions (regions within Bolivia or other countries), while label shape indicates growth habit (determinate, semi-determinate, indeterminate).

**Figure 6 genes-13-01505-f006:**
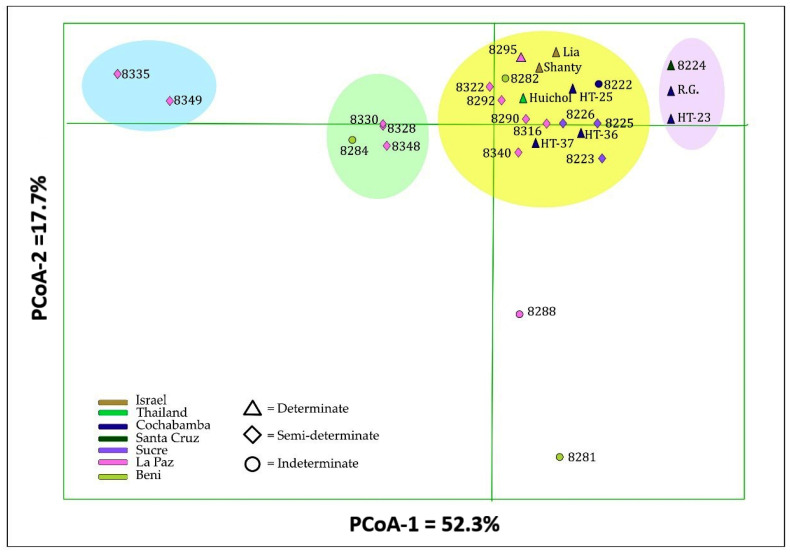
Principal coordinate analysis (PCoA) bi-plot, generated based on Nei’s unbiased genetic distance, demonstrating the relationship between the 28 tomato accessions, with the first two principal coordinates (PCoA1 and PCoA2) explaining 70% of the total variation. Accessions with the same label colour belong to the same region within Bolivia, or to the same country.

**Figure 7 genes-13-01505-f007:**
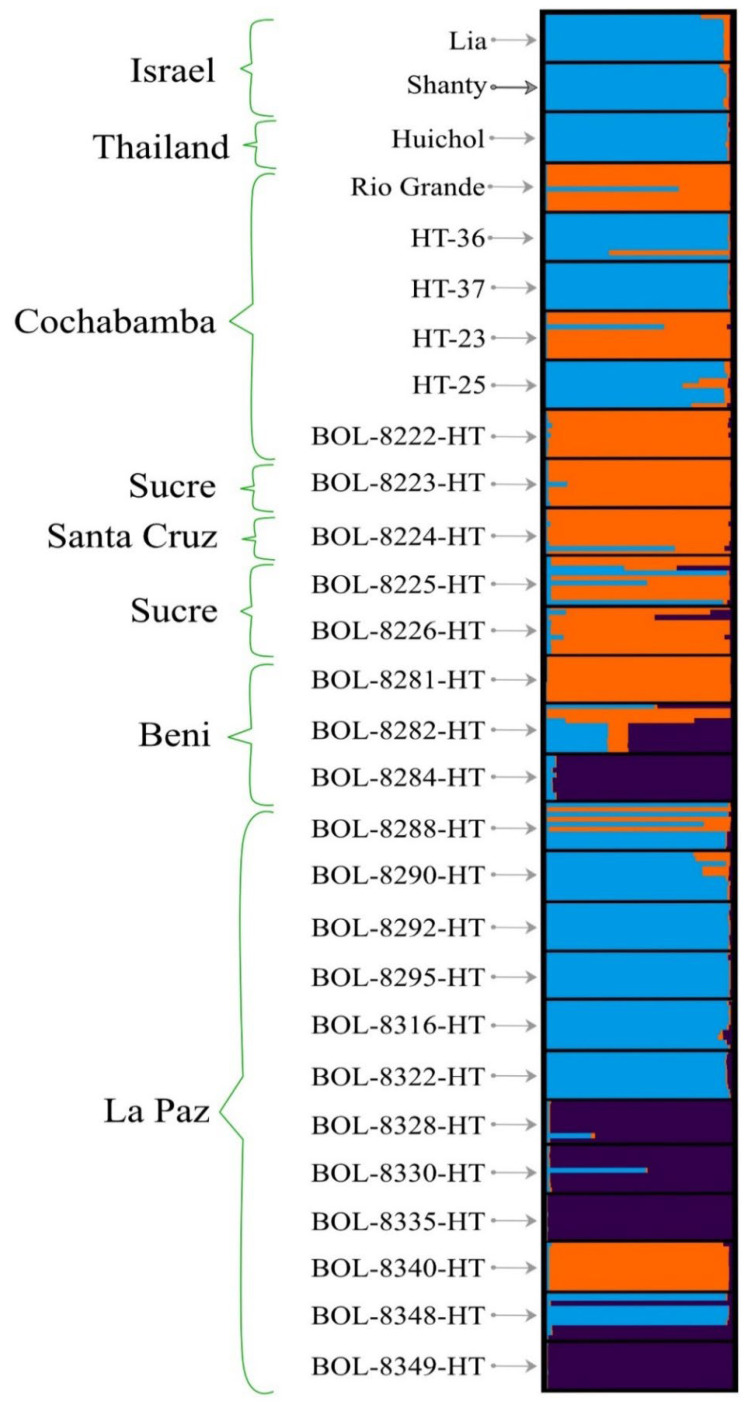
Graphical display of optimal genetic structure of the 279 individual genotypes representing the 28 tomato accessions. Light blue, yellow and deep purple represent the three clusters (K) identified in population structure analysis. In each accession, the proportion of each colour represents the average proportion of alleles that placed each accession in one or more cluster. A black rectangular border delimits each accession.

**Table 1 genes-13-01505-t001:** Description of the 28 tomato accessions used in this study.

Accession/Commercial Name	Germ-Plasm Provider	Country of Origin	Region of Sampling Site in Bolivia	Domestication/Breeding Status	Geographical Position of Sampling Site	Altitude of Sampling Site (Masl)	Fruit Shape	Fruit Size	Fruit Colour	Plant Flowering Type
Lia	Hazera	Israel	--	Cultivated	Not applicable	Not applicable	Cylindrical	Intermediate	Red	Determinate
Shanty	Hazera	Israel	--	Cultivated	Not applicable	Not applicable	Cylindrical	Intermediate	Red	Determinate
Huichol	Seminis	Thailand	--	Cultivated	Not applicable	Not applicable	Cylindrical	Intermediate	Red	Determinate
Rio Grande	CNPSH	Bolivia	Cochabamba	Cultivated	17°26′24″ S; 66°20′47″ W	2548	Cylindrical	Intermediate	Red	Determinate
HT-36	CNPSH	Bolivia	Cochabamba	Advanced line	17°26′24″ S; 66°20′47″ W	2548	Rounded	Very large	Red	Determinate
HT-37	CNPSH	Bolivia	Cochabamba	Advanced line	17°26′24″ S; 66°20′47″ W	2548	Rounded	Large	Red	Determinate
HT-23	CNPSH	Bolivia	Cochabamba	Advanced line	17°26′24″ S; 66°20′7″ W	2548	Cylindrical	Intermediate	Red	Determinate
HT-25	CNPSH	Bolivia	Cochabamba	Advanced line	17°26′24″ S; 66°20′47″ W	2548	Cylindrical	Intermediate	Red	Determinate
BOL-8222-HT	BGH-BNG	Bolivia	Cochabamba	Cultivated	17°23′03″ S; 66°08′05″ W	2858	High rounded	Intermediate	Red	Determinate
BOL-8223-HT	BGH-BNG	Bolivia	Sucre	Cultivated	19°17′43″ S; 64°22′33″ W	2201	High rounded	Very small	Yellow	Semi-determinate
BOL-8224-HT	BGH-BNG	Bolivia	Santa Cruz	Cultivated	17°24′00″ S; 63°53′00″ W	300	High rounded	Small, intermediate	Red	Determinate
BOL-8225-HT	BGH-BNG	Bolivia	Sucre	Cultivated	19°44′26″ S; 63°52′24″ W	1165	Slightly flattened	Very small	Pink	Semi-determinate
BOL-8226-HT	BGH-BNG	Bolivia	Sucre	Cultivated	19°48′26″ S; 64°00′29″ W	1143	Rounded	Very small	Yellow	Semi-determinate
BOL-8281-HT	BGH-BNG	Bolivia	Beni	Wild	14°52′10.7″ S; 61°04′42.3″ W	227	Rounded	Very small	Red	Indeterminate
BOL-8282-HT	BGH-BNG	Bolivia	Beni	Cultivated	Not reported	227	Slightly flattened	Small	Red	Indeterminate
BOL-8284-HT	BGH-BNG	Bolivia	Beni	Cultivated	15°08′47.4″ S; 61°02′15″ W	259	Slightly flattened	Very small	Red	Indeterminate
BOL-8288-HT	BGH-BNG	Bolivia	La Paz	Wild	15°47′32″ S; 60°58′41″ W	498	Rounded	Very small	Red	Indeterminate
BOL-8290-HT	BGH-BNG	Bolivia	La Paz	Wild	15°48′28″ S; 61°37′27″ W	594	Slightly flattened	Very small	Yellow	Semi-determinate
BOL-8292-HT	BGH-BNG	Bolivia	La Paz	Wild	16°15′47″ S; 61°41′44″ W	1676	Slightly flattened	Very small	Yellow	Semi-determinate
BOL-8295-HT	BGH-BNG	Bolivia	La Paz	Wild	16°11′21.6″ S; 67°43′29″ W	599	Slightly flattened	Small, Intermediate	Red	Determinate
BOL-8316-HT	BGH-BNG	Bolivia	La Paz	Cultivated	15°58′12″ S; 67°27′44″ W	961–1030	Slightly flattened	Intermediate	Red	Semi-determinate
BOL-8322-HT	BGH-BNG	Bolivia	La Paz	Cultivated	16°11′53″ S; 67°42′16″ W	1725–1690	Slightly flattened	Very small	Red	Semi-determinate
BOL-8328-HT	BGH-BNG	Bolivia	La Paz	Wild	16°15′31″ S; 67°41′32″ W	1853–1870	Flattened	Very small	Red	Semi-determinate
BOL-8330-HT	BGH-BNG	Bolivia	La Paz	Wild	16°11′19″ S; 67°43′29″ W	1716–1720	Slightly flattened	Very small	Red	Semi-determinate
BOL-8335-HT	BGH-BNG	Bolivia	La Paz	Wild	16°20′21″ S; 67°26′38″ W	1124–1190	Slightly flattened	Very small	Red	Semi-determinate
BOL-8340-HT	BGH-BNG	Bolivia	La Paz	Wild	16°26′09″ S; 67°28′26″ W	1492–1550	Slightly flattened	Very small	Red	Semi-determinate
BOL-8348-HT	BGH-BNG	Bolivia	La Paz	Wild	16°28′35″ S; 67°26′59″ W	2021–2012	Rounded	Very small	Yellow	Semi-determinate
BOL-8349-HT	BGH-BNG	Bolivia	La Paz	Wild	16°28′35″ S; 67°26′59″ W	2021–2010	Rounded	Very small	Red	Semi-determinate

Note: Accession description sources can be retrieved without suffixes BOL-HT from the website http://germoplasma.iniaf.gob.bo/ (accessed on 27 July 2022).

**Table 3 genes-13-01505-t003:** Estimates of different population genetics parameters for the 28 tomato accessions studied.

Genotype	Na	Ne	NPA	NLCA ≤ 0.25	NLCA ≤ 0.50	PPL	I	Ho	He	uHe	F
‘Lia’	1.18	1.18	0.00	0.09	0.27	0.18	0.13	0.18	0.09	0.10	−1.00
‘Shanty’	1.27	1.27	0.00	0.18	0.27	0.27	0.19	0.27	0.14	0.15	−1.00
‘Huichol’	1.27	1.21	0.00	0.09	0.27	0.27	0.16	0.18	0.11	0.12	−0.33
‘Rio Grande’	1.36	1.05	0.00	0.27	0.46	0.36	0.08	0.03	0.04	0.04	0.21
‘HT-36′	1.09	1.09	0.00	0.00	0.18	0.09	0.06	0.09	0.05	0.05	−1.00
‘HT-37’	1.18	1.03	0.00	0.00	0.18	0.18	0.05	0.01	0.03	0.03	0.47
‘HT-23’	1.18	1.12	0.00	0.27	0.36	0.09	0.09	0.00	0.05	0.06	1.00
‘HT-25’	1.64	1.34	0.00	0.27	0.64	0.55	0.29	0.24	0.18	0.20	−0.17
BOL-8222-HT	1.36	1.24	0.09	0.18	0.36	0.27	0.18	0.05	0.11	0.12	0.52
BOL-8223-HT	1.00	1.00	0.00	0.09	0.18	0	0.00	0.00	0.00	0.00	na
BOL-8224-HT	1.09	1.03	0.00	0.18	0.36	0.09	0.03	0.02	0.02	0.02	−0.14
BOL-8225-HT	1.18	1.07	0.09	0.18	0.27	0.09	0.07	0.00	0.04	0.04	1.00
BOL-8226-HT	1.36	1.25	0.00	0.27	0.46	0.36	0.22	0.03	0.15	0.18	0.70
BOL-8281-HT	1.00	1.00	0.00	0.36	0.36	0	0.00	0.00	0.00	0.00	na
BOL-8282-HT	1.18	1.07	0.09	0.27	0.36	0.18	0.08	0.00	0.05	0.06	1.00
BOL-8284-HT	1.00	1.00	0.00	0.18	0.36	0	0.00	0.00	0.00	0.00	na
BOL-8288-HT	1.46	1.38	0.00	0.46	0.64	0.45	0.29	0.03	0.21	0.25	0.87
BOL-8290-HT	1.09	1.06	0.00	0.00	0.09	0.09	0.05	0.00	0.03	0.04	1.00
BOL-8292-HT	1.36	1.18	0.00	0.18	0.36	0.36	0.18	0.03	0.12	0.13	0.81
BOL-8295-HT	1.09	1.05	0.00	0.00	0.18	0.09	0.05	0.01	0.03	0.03	0.58
BOL-8316-HT	1.18	1.15	0.00	0.00	0.18	0.18	0.12	0.00	0.08	0.10	1.00
BOL-8322-HT	1.27	1.12	0.00	0.00	0.27	0.27	0.13	0.00	0.08	0.09	1.00
BOL-8328-HT	1.00	1.00	0.00	0.09	0.18	0	0.00	0.00	0.00	0.00	na
BOL-8330-HT	1.00	1.00	0.00	0.09	0.18	0	0.00	0.00	0.00	0.00	na
BOL-8335-HT	1.27	1.08	0.09	0.27	0.36	0.27	0.11	0.04	0.06	0.07	0.26
BOL-8340-HT	1.00	1.00	0.00	0.18	0.18	0	0.00	0.00	0.00	0.00	na
BOL-8348-HT	1.55	1.37	0.09	0.36	0.55	0.45	0.29	0.04	0.19	0.20	0.81
BOL-8349-HT	1.09	1.01	0.00	0.36	0.46	0.09	0.02	0.01	0.01	0.01	−0.07
	1.21	1.12	0.02	0.18	0.32	0.19	0.10	0.05	0.07	0.07	0.36
	0.03	0.02	0.02	0.10	0.15	0.03	0.01	0.01	0.01	0.01	0.04

Na = Observed number of alleles; Ne = Effective number of alleles; NPL = Number of private alleles (number of alleles unique to a single population); NLCA ≤ 0.25 = Number of locally common alleles found in 25% or fewer accessions; NLCA ≤ 0.50 = Number of locally common alleles found in 50% or fewer accessions; PPL = Percentage of polymorphic loci; I = Shannon’s information index; He = Expected heterozygosity; Ho = Observed heterozygosity; uHe = Unbiased expected heterozygosity; F = Fixation index. Note: Loci with private alleles in populations BOL-8222-HT, BOL-8225-HT, BOL-8282-HT, BOL-8335-HT are SSR22, SLM6-11, LE20592, TomSatX2-2 and TomSatX11-1, respectively.

**Table 4 genes-13-01505-t004:** Results of analysis of molecular variance (AMOVA) based on 1000 permutations without grouping the accessions and on grouping them according to geographical region of origin, altitude, cultivation status, fruit shape, fruit colour, fruit size and growth type.

Grouping Factor	Source of Variation	Degrees of Freedom	Sum of Squares	Variance Components	Percentage of Variation	Fixation Indices	Probability (P) Value
	Among accessions	27	464.92	1.349 Va	77.29	F_ST_ = 0.77	Va & F_ST_ = 0.000
	AIWA *	146	75.86	0.123 Vb	7.07	F_IS_ = 0.31	Vb & F_IS_ = 0.000
	Within individuals	174	47.50	0.273 Vc	15.64	F_IT_ = 0.84	Vc & F_IT_ = 0.000
	Total	347	588.28	1.745			
Geographical	^a^ Among groups	1	83.48	0.438 Va	21.71	F_CT_ = 0.22	Va & F_CT_ = 0.000
region of origin	AAWGr **	23	351.54	1.232 Vb	61.07	F_SC_ = 0.78	Vb & F_SC_ = 0.000
	Within accessions	281	97.61	0.347 Vc	17.22	F_ST_ = 0.82	Vc & F_ST_ = 0.000
	Total	305	532.63	2.018			
Altitude	^b^ Among groups	3	54.43	−0.027 Va	−1.43	F_CT_ = −0.01	Va & F_CT_ = 0.566
groups	AAWGr	15	260.65	1.602 Vb	85.08	F_SC_ = 0.84	Vb & F_SC_ = 0.000
	Within accessions	189	58.19	0.308 Vc	16.35	F_ST_ = 0.84	Vc & F_ST_ = 0.000
	Total	207	373.27	1.883			
Cultivation	^c^ Among groups	1	61.93	0.361 Va	17.92	F_CT_ = 0.18	Va & F_CT_ = 0.005
status	AAWGr	21	332.80	1.276 Vb	63.36	F_SC_ = 077	Vb & F_SC_ = 0.000
	Within accessions	259	97.61	0.377 Vc	18.72	F_ST_ = 0.81	Vc & F_ST_ = 0.000
	Total	281	492.34	2.013			
Fruit shape	^d^ Among groups	2	71.59	0.282 Va	16.75	F_CT_ = 0.16	Va & F_CT_ = 0.000
	AAWGr	16	195.54	0.948 Vb	56.25	F_SC_ = 0.67	Vb & F_SC_ = 0.000
	Within accessions	221	100.56	0.455 Vc	26.99	F_ST_ = 0.73	Vc & F_ST_ = 0.000
	Total	239	367.69	1.686			
Fruit colour	^e^ Among groups	2	12.73	−0.034 Va	−2.05	F_CT_ = −0.02	Va & F_CT_ = 0.533
	AAWGr	24	378.92	1.263 Vb	77.11	F_SC_ = 0.76	Vb & F_SC_ = 0.000
	Within accessions	292	119.28	0.408 Vc	24.94	F_ST_ = 0.75	Vc & F_ST_ = 0.000
	Total	317	510.93	1.634			
Fruit size	^f^ Among groups	1	70.74	0.440 Va	22.53	F_CT_ = 0.23	Va & F_CT_ = 0.000
	AAWGr	19	252.18	1.076 Vb	55.08	F_SC_ = 0.71	Vb & F_SC_ = 0.000
	Within accessions	233	101.97	0.438 Vc	22.39	F_ST_ = 0.77	Vc & F_ST_ = 0.000
	Total	253	424.89	1.954			
Growth type	^g^ Among groups	2	111.77	0.420 Va	21.98	F_CT_ = 0.22	Va & F_CT_ = 0.000
	AAWGr	25	353.13	1.106 Vb	57.85	F_SC_ = 0.74	Vb & F_SC_ = 0.000
	Within accessions	320	123.36	0.385 Vc	20.16	F_ST_ = 0.79	Vc & F_ST_ = 0.000
	Total	347	588.28	1.912			

* AIWA = among individuals within accessions; ** AAWGr = among accessions within groups. ^a^ The 25 Bolivian accessions were divided into two groups based on their geographical region of origin (La Paz vs. Cochabamba + Chuquisaca + Santa Cruz + Beni); ^b^ Nineteen Bolivian accessions with known altitude of collecting site were divided into four altitude groups (<500 masl, 950–1200 masl, 1450–1750 masl and 1850–2250 masl); ^c^ Twenty-three Bolivian accessions with known cultivation status were divided into two groups (cultivated and wild); ^d^ Nineteen accessions were divided into three groups according to fruit shape (cylindrical, round and slightly flattened); ^e^ Twenty-six accessions were divided into two groups according to fruit colour (red vs. yellow); ^f^ Twenty-one accessions were grouped into two groups according to fruit size (intermediate vs. very small); ^g^ Twenty-eight accessions were divided into three groups according to growth type (determinate, semi-determinate and indeterminate).

## Data Availability

Data is stored at the SLU server and available by contact with the first author.

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
