# Peer review of "Simple Sequence Repeat Markers Reveal Genetic Diversity and Population Structure of Bolivian Wild and Cultivated Tomatoes (Solanum lycopersicum L.)"

_genes, 2022, doi:10.3390/genes13091505_

Round 1

Reviewer 1 Report

This paper evaluated the genetic diversity and population structure of  28 Bolivian tomato accessions, and relate their genetic variation. Germplasm evaluation is important before undertaking breeding activities. There are some suggestions as below:

1: The authors need to address the current progress of germplasm evaluation on origin, classfication, and genetic diversity as well  in tamato.

2: As the 28 accessions are not a big number of accessions, the names and the distribution information are important for the other public authors if the paper published. I suggest these information (Suplemmentary Table1, Figure S-3, Figure S-1) could be included in the article, rather than just as the Suplementary materials.

3: Try to use subtitles in the part of Discussion, so that it could be read more clearly.

Reviewer 2 Report

The manuscript demonstrates genetic relationship of cultivar and wild population of Bolivian tomato and its genetic diversity. Number of samples and statistical analysis method is appropriate.

Comments

CBD Nagoya protocol issues: Bolivian tomatoes are grown in Sweden. Authors should demonstrate or declare compliance of Nagoya protocol of the Convention of Biological Diversity.

Keywords: "Core germplasm":

1) Although authors mentioned various times regarding "Bolivian core collection", there are no detailed description of how the "Bolivian core collection" were selected. I could not access any of mentioned web sites appear on the reference list. Authors should provide additional description of how Bolivian core collection were selected in introduction.

2) Authors chose "Core germplasm" as the first key words, but there is no discussion on the "Core germplasm" in the discussion part. Add some discussion about it or reconsider the key word.

3) Number of accessions from La Paz region are considerably larger (12) compared with materials from other regions (6 from Cochabamba, 3 from Beni, 3 from Sucre, 1 from Santa Cruz). Since no information provided how this "Bolivian core germplasm" were selected, the conclusion "La Paz germplasm plays a significant role in tomato improvement programs in Bolivia" (LINE 637) is not much persuasive.

Minor correction

LINE 66: core-collection >> core collection (other places do not include "-" between core and collection) .

LINE 193: as well as fixation indices. >> as well as fixation indices (F).

Table 3: A column indicates characteristics (e.g. geographic regions of origin, altitudinal group etc.) should be added as the first column of Table 3.

References: Please provide access date for the web references. I couldn't access to the website of [6] and [8].
